# Plasma-Induced Bubble Microjet Metallization of Elastomer

**DOI:** 10.3390/mi10060389

**Published:** 2019-06-11

**Authors:** Keita Ichikawa, Natsumi Basaki, Yu Yamashita, Yoko Yamanishi

**Affiliations:** Department of Engineering, Kyushu University, Fukuoka 819-0395, Japan; 3TE18662G@s.kyushu-u.ac.jp (K.I.); 2TE19694K@s.kyushu-u.ac.jp (N.B.); 2TE18714S@s.kyushu-u.ac.jp (Y.Y.)

**Keywords:** nickel, nanoparticles, wiring, adhesion, latex rubber

## Abstract

As the development of flexible materials and advanced materials progresses, innovative wiring methods for these materials are attracting attention. In this study, we investigated a new wiring technology using plasma-induced microbubbles for elastomer without any surface treatment. Our technology includes three main points. (1) Unlike electroless plating and other conventional methods, it does not require complicated pre-surface treatment processes before wiring. (2) A wiring resolution of 500 micro meter can be reached quickly and economically. (3) Robust metallic adhesion on a wide range of materials can be successfully carried out with precise positioning. Here, by applying our method, we adhered nickel nanoparticles to a latex rubber substrate and demonstrated the electrical conductivity of the created line. The result suggests that our method has potential as an innovative wiring technology to precisely, robustly, and simply fabricate an electric circuit without any complicated procedures or pre-treatment. Our method can contribute to microfabrication technologies.

## 1. Introduction

Many studies have employed various modeling technologies, e.g., three-dimensional printing, to realize lightweight and high-strength advanced materials, such as carbon fiber reinforced plastics (CFRP) [1,2,3]. Three-dimensional printing has enabled the economical and rapid modeling of complex shapes with diverse materials. In many fields, advanced materials have shown advantages such as weight reduction and energy saving. Furthermore, fabricating electric circuits on advanced materials and 3D printed objects adds new value as electronic components such as sensors.

Generally, 3D-printed objects and advanced materials composed of fibers and elastomers are almost non-conductive. Electroless plating has been used as a wiring method for non-conductive materials. The electroless plating process involves the following steps. First, the surface of the target object is pre-treated with a specific reagent to catalyze the surface. Second, the treated object is immersed in a plating solution, and a redox reaction starts between the metal ions present in the solution and the catalyzed surface. Finally, the reduced metal is deposited on the object’s surface [4]. Although the electroless plating method can be applied to various materials, the required surface treatment has some issues such as a very complicated, long process and the consumption of a large amount of plating solution [5].

This study aimed to develop an innovative wiring method for various materials and shapes without pre-surface treatment using a simple process. We present a wiring method that uses discharging and plasma-induced bubbles to adhere metallic particles to the surface. Herein we report the optimization of particle adhesion conditions and the wiring results for rubber.

## 2. Device and Concept of the Method

### 2.1. Bubble Generation Device

We developed a device that was able to generate and eject microbubbles using a local high electric field in an electrolyte solution. Figure 1 shows the structure of our device. By applying a pulsed high voltage to the electrode, a high electric field was generated in the hole part at the tip of the device. 

Figure 2 shows the result of the finite elemental method (FEM) analysis of the electric field when a voltage of 1500 V was applied to the electrode. A high electric field of several MV/m was generated in the hole. Moreover, the gas that was present in the solution caused a dielectric breakdown due to the high electric field, generating plasma [6]. It is generally known that a microbubble grows rapidly due to a dielectric breakdown in the solution. Hence, a microbubble was generated rapidly at the tip of the device [7,8]. Furthermore, the generated microbubble caused an inhomogeneous collapse, forming a jet flow called a microjet. The microjet has an extremely high pressure. It can damage hard materials such as metals [9,10]. Figure 3 shows sequence photographs from bubble generation to ejection.

### 2.2. Concept of the Wiring Method

Figure 4 shows the concept and process of our method. We used an electrolyte solution containing metal particles. Applying a pulsed voltage to the device in the solution generated microbubbles, which were ejected onto the target. The ejected microbubbles caused an inhomogeneous collapse, and forming a microjet that involved the surrounding metal particles. The microjet scarred the target surface and adhered metal particles to the surface. Finally, one point was metalized. By continuously performing this process, metal wires were formed. Our method was able to hold the created wire on the surface of the target by adhesion of the metal particles on the surface.

## 3. Optimization Experiment of the Distance for the Output Power

In this experiment, the applied output of the power supply was fixed. We aimed to clarify the particle deposition trend when the distance between the device and a target object changed. Finally, we determined the optimum distance at a fixed output condition.

### 3.1. Experimental Setup

In the present method, the distance between the device and the target greatly influenced the force to scar some of the target’s surface and adhere metal particles to the scarred point. We fixed the conditions of the output power and the solution, and then we optimized the distance L, which is that between the device tip and the surface of the target.

Figure 5 shows the experimental setup. The target was a PVC (polyvinyl chloride) board, and the solution was a mixture of nickel particles (diameter D = 100 nm) and a 0.9% sodium chloride solution. The power supplier was a high-frequency power supply for an electric scalpel. The output power and the number of output times were fixed at 15 W and 30 times, respectively. The distance L was controlled by a precision manipulator, and particles were adhered every 100 µm between L = 0 to 1 mm. After the experiment, we measured the diameter of the metallized area and the burnt area by discharging. The experiment was performed three times at each distance, and the diameter of the deposited area was measured every time. We used the average value and the maximum burnt diameter for the results.

### 3.2. Results

Figure 6 shows the deposited diameter of the nickel particles and the burnt diameter at each distance. When the distance L = 500 µm, the deposition diameter peaked and the deposition diameter decreased as the distance deviated from L = 500 µm (Figure 7). Furthermore, as the deposition area increased, the density of the nickel particles (black area) also increased. On the other hand, when the distance L = 0, the diameter of the burnt area peaked due to discharging. However, the burnt area decreased exponentially as the distance L increased. The reason the deposition diameter was maximized at the distance L = 500 µm was that the distance between the device and the target agreed with the conditions to realize the maximum pressure of the generated microjet. It is generally known that the pressure of a microjet generated from a microbubble depends on the distance between the target and the generated bubbles, reaching maximum pressure at a specific distance. For microbubbles generated under the conditions in this study, the pressure was maximized at a distance L = 500 µm.

When wiring with the metal particles, the particle density was the most important parameter. Therefore, under the conditions in this study, the distance L = 500 µm was the optimum condition.

## 4. Wiring Experiment

### 4.1. Experimental Setup

In this experiment, we used the same device, power supply, and solution as in the previous experiment. As the target, we used latex rubber, which is a flexible material. The process of this experiment was as follows. First, we added tension to the rubber with two weights in order to fix the rubber in a 200% extended state. Then, we created a droplet with a solution containing the nickel particles. Next, we moved the device and the opposite electrode with the precision manipulator and inserted them into the droplet. After that, we adjusted the distance between the device and the rubber to 500 µm and adhered particles to one point by applying the pulsed voltage from the power supply. Then, we moved the device 200 µm using the manipulator and applied the voltage again. Wiring was completed by repeating this process. Finally, we removed the weights and returned the rubber to its original length. The particle density was able to increase by stretching the rubber before this experiment and returned to its original length after the experiment (Figure 8).

After the experiment, we connected a copper wire to the created line to realize an electric circuit and confirm the energization. In addition, we evaluated adhesion by a peel test using sellotape, which is the standard of adhesion for a general electric circuit.

### 4.2. Results

Figure 9 shows magnified images of the latex rubber after wiring with and without tension. This time, the maximum moving speed of the device was 30 µm/s and the bubble generating frequency was 450 kHz. The image in Figure 9 was taken after the peel test and ultrasonic cleaning. The created line remained in the rubber without delamination". Therefore, by our method, the created wire had the same adhesion as a general electric circuit. Moreover, wiring on the rubber in the extended state prevented wire breakage when the substrate elongated and contracted. Figure 10 shows a scanning electron microscope (SEM) image of the created line. 

Figure 11 shows the electric circuit confirming the energization using light-emitting diode (LED). We successfully demonstrated light emission of the LED. Accordingly, the created line by the present method had conductivity.

### 4.3. Discussion

First, we considered the appropriate material in this method. The adhesion between the particles and the target surface was mainly electrostatic. In addition, it was preferable for the substrate to be stretchable in order to improve the attached density. Therefore, as a material of the substrate, an easily charged and stretchable insulating material was preferable. 

Table 1 summarizes some characteristics of our method and the other developed method. In our method, the device movement speed was 30 μm /s. On the other hand, in the method using inkjet and plasma jet, device movement speed was 1.2 to 4 m/s, confirming that the moving speed of our method was very low. However, in the other method, the heat treatment took a very long time (30 min to 1 h). On the other hand, our method had almost no heat treatment time. Therefore, comparing the total process time, our method could complete wiring in a shorter time when wiring below 40 m (our device is able to move 40 m in 30 min, using our method). 

The wiring resolution was very low at 500 µm in this experiment. However, it was found that the resolution was greatly influenced by the tip shape of the bubble injector, especially the hole diameter of the bubble injector. At present, the resolution is expected to be improved to about 10 to 30 µm. Moreover, we are currently working on bubble injector redesign which promises to improve resolution significantly.

## 5. Conclusions

This study successfully developed a new wiring method for non-conductive materials. Here, a method to adhere metal particles using plasma-induced bubbles was presented. The tendencies of the particle deposition diameter against the distance between the device and the target using nickel particles were confirmed. In addition, we made a line on the rubber with the present method and demonstrated the energization of the created line with the emission of the LED. The result suggested that the present method has the potential to become a new wiring method for various materials.

In future research, we will measure the volume resistivity of the created line and compare the resistance with conventional wiring methods. In addition, we will evaluate the effects of particle deposition and oxidation by measuring cross-sectional images of the adhesion particles by transmission electron microscope (TEM) and the particle surface conditions by X-ray photoelectron spectroscopy (XPS). 

## Figures and Tables

**Figure 1 micromachines-10-00389-f001:**
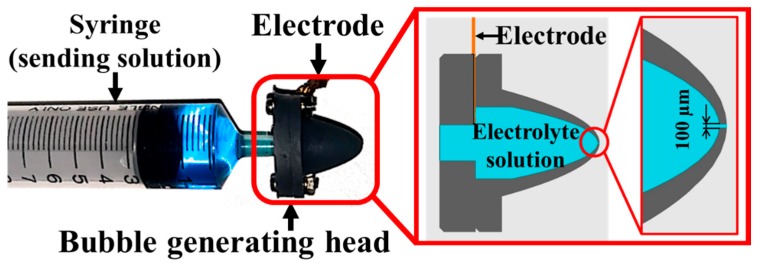
Overall view and structure of the device, which is composed of a syringe and a bubble generating head. The head was made with a 3D printer. It has a small hole at the tip and a hollow space inside. The space is filled with an electrolyte solution and the electrode is in contact with the solution.

**Figure 2 micromachines-10-00389-f002:**
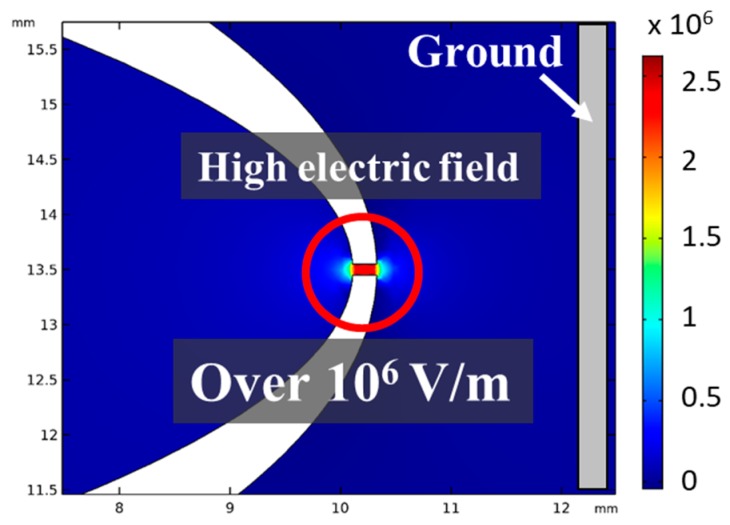
Result of the finite elemental method (FEM) analysis of the electric field (applied voltage = 1500 V). It was confirmed that a high electric field over 10^6^ V/m was generated at the small hall part of the bubble injector. The value of the generated electric field was over the dielectric strength of the present gas in the solution and can cause a dielectric breakdown and generate plasma.

**Figure 3 micromachines-10-00389-f003:**
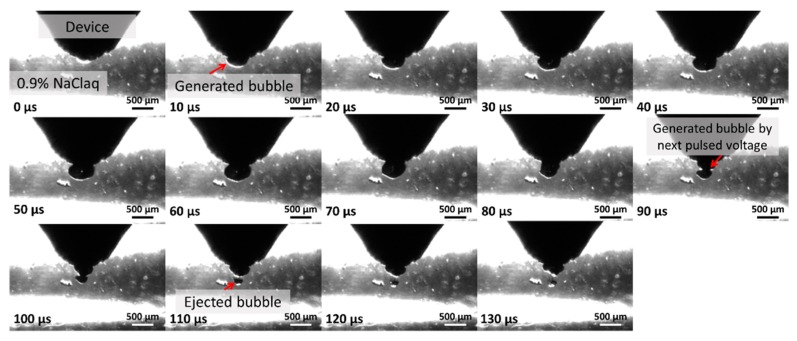
Sequence photographs of the bubble generation and ejection taken at 500,000 fps by an ultra-high-speed camera (HPV-X2, SHIMADZU, Kyoto, Japan).

**Figure 4 micromachines-10-00389-f004:**
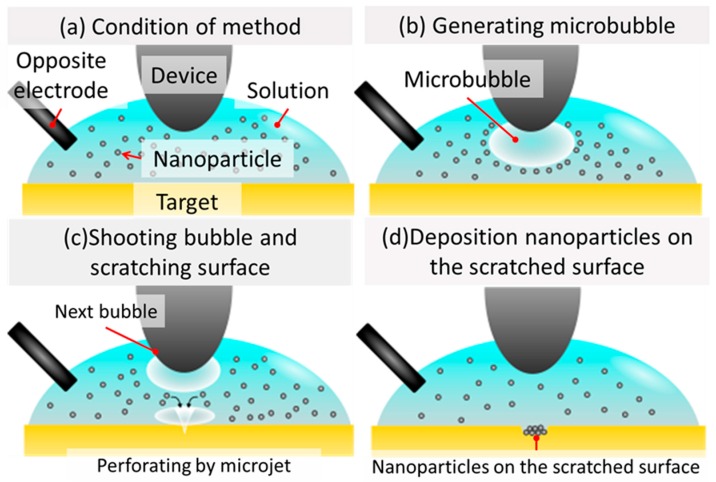
Concept and process flow of the presented method. (**a**) The bubble injector and opposite electrode were inserted into the solution, and the nanoparticles were dispersed in the solution. (**b**) Pulsed voltage was applied to the device, and a microbubble was generated in the solution. (**c**) The generated bubble was shot and scratched the surface of the target. (**d**) The nanoparticles were deposited and adhered to the part of the scratched surface of the target.

**Figure 5 micromachines-10-00389-f005:**
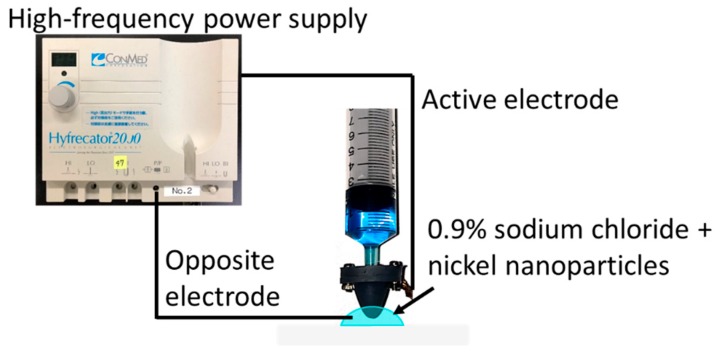
Experimental setup to optimize the distance for the output power. The power supply was connected to the bubble injector and an electric circuit was created through the opposite electrode immersed in the solution.

**Figure 6 micromachines-10-00389-f006:**
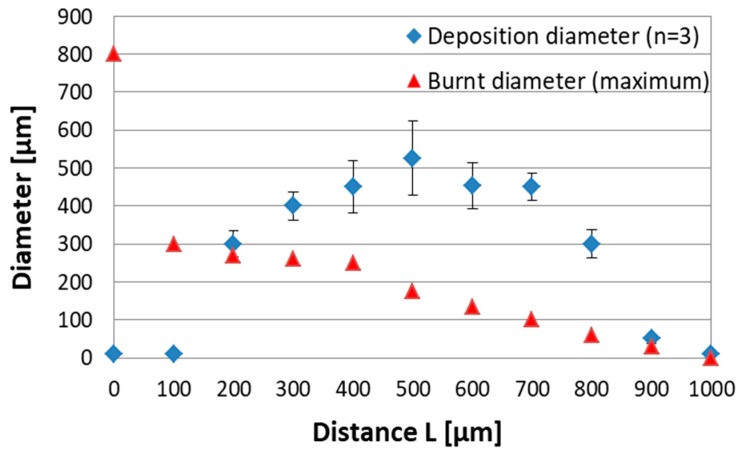
Measurement results of deposited particles and burnt diameter at each distance. The deposition diameter was maximized at the distance L = 500 µm, whereas that for the burnt diameter was at the distance L = 0.

**Figure 7 micromachines-10-00389-f007:**
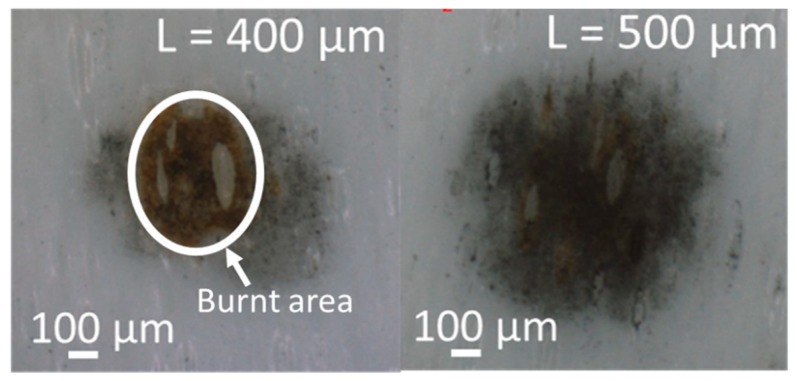
Example of an experimental result for (left) distance L = 400 µm and (right) distance L = 500 µm. The density of the nickel particles (black area) increased as the deposition diameter was enlarged.

**Figure 8 micromachines-10-00389-f008:**
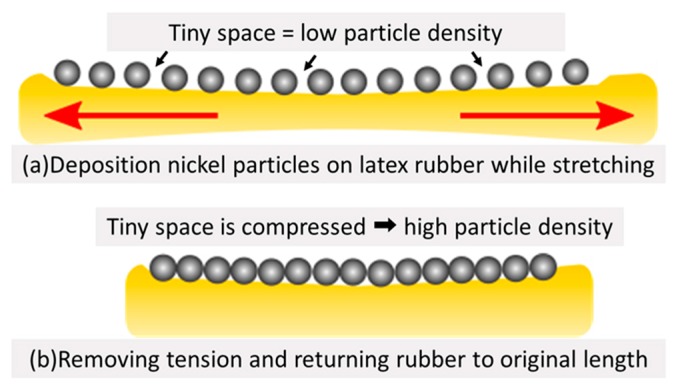
Concept and content of the extending latex rubber. (**a**) When the particles were adhered in an extended state, there was the possibility that a tiny space existed between the adhesion particles, causing low particle density. (**b**) By returning the rubber to its original length, the tiny space was compressed and the particle density increased.

**Figure 9 micromachines-10-00389-f009:**
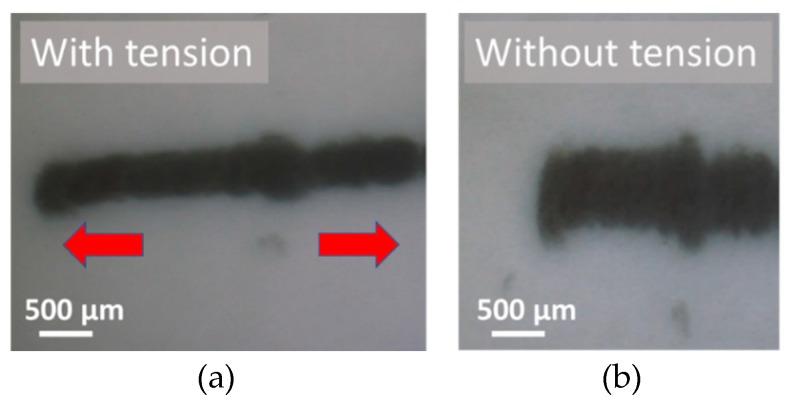
Result of the wiring experiment. (**a**) Magnified view of the rubber with tension and (**b**) without tension.

**Figure 10 micromachines-10-00389-f010:**
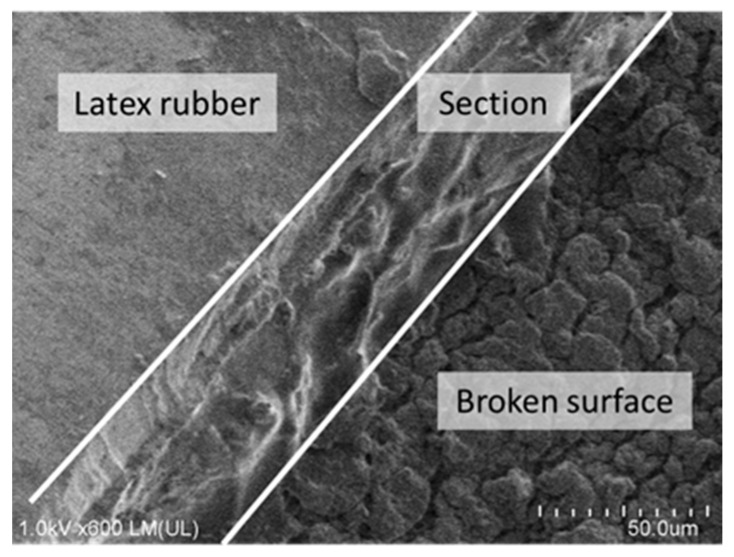
Scanning electron microscope (SEM)image of the line created with the bubble injector. The left area shows the surface of the latex rubber, the center area shows the surface of the cross-section, and the right area shows the broken surface of the latex rubber caused by the bubble injector.

**Figure 11 micromachines-10-00389-f011:**
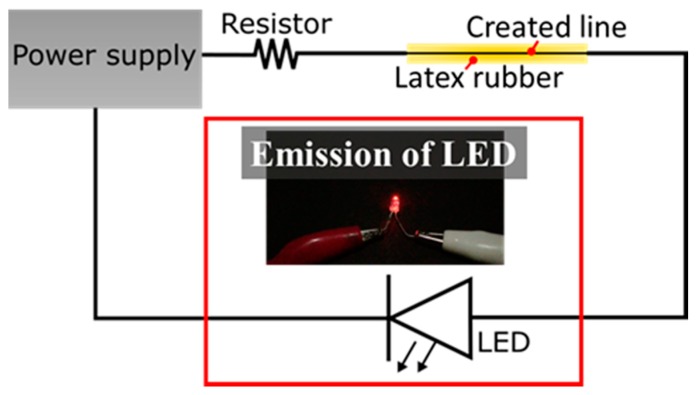
Electrical schematic to confirm the energization of the created line using LED. The LED was connected to the power supply via a resistor (3.3 kΩ) and the created line on the latex rubber.

**Table 1 micromachines-10-00389-t001:** Summary of the characteristics of our method and the other developed method.

Method	Device Movement Speed	Resolution	Heat Treatment	Maximum Temperature	Total Process Time
Our Method	30 μm/s	500 μm	-	room temperature	few min
Inkjet [11]	1.2 m/s	45.7 ± 0.5 μm	30 min	400 °C	over 30 min
Inkjet [12]	3.8–3.9 m/s	65 μm	1 h	350 °C	over 1 h
Plasma Jet [13]	4 m/s	250 μm	35 min	190 °C	over 35 min

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
