# Peer review of "Plasma-Induced Bubble Microjet Metallization of Elastomer"

_micromachines, 2019, doi:10.3390/mi10060389_

Round 1
Reviewer 1 Report
First I would like to argue about the title:
"Plasma-induced bubble microjet metallization of elastomer"
would bring focus to the most important aspects of the paper.
The original is also wrong in the sense that "various materials" were not studied,
only one type of rubber.
If the title is changed as prposed, I would use the following keywords:
nickel
nanoparticles
wiring
adhesion
latex rubber.
It is useless to repeat title words in keywords.
The main problem of the manuscript as I see it is lack of numerical data.
At least some data on resistance of wiring should be reported, and compared with e.g. electroplated nickel of similar dimensions.
Figure 4 is difficult to read: I would say that a simple 2D cross section figure would serve better.
In figure embedded nanoparticles are mentioned but the main text just above uses the term "implanting the metal particles on the surface." Implanting and embedding both imply that particles will go inside the elatomer, with "on the surface" indicates that they accumulate on top of the material. Therefore, cross sectional figure 4 would be useful to actually show where the particles are placed: inside material or on surface.
Some effort must be taken to examine the depth of nickel layer.
Strain of the rubber must be written out: how many % extension during metal deposition ?
Figure 8 uses too much space to repeat the overall system set-up (compeare Fig. 4). and should have more emphasis on stretching and particle deposition.
It would be interesting if scaling of this method was discussed: 500 um linewidth is rather conservative and will limit potential applications.
The manuscript lacks an important part: comparison to literature. Some competing methods should be discussed and results compared. For example: writing speed, minimum linewidth, metal resistivity, metal adhesion etc. Some methods are probably better in some aspects and worse in some, but it is the job of the authors to do this comparison, so that readers can evaluate if the new method offers advantages over existing methods. This literature analysis also helps to bring the focus of the article clear, when shortcomings of old methods and benefits of new method are discussed side-by-side.
Additional comments:
Nickel particles are probably 100 nm and not 100 um ?!
Non-inductive resistor ?
Fig. 10: I would prefer an electrical circuit schematic, with a photo of LED.
Author Response
To the editor and reviewers of Journal of micromachines
Subject: Submission of manuscript entitled “Plasma-induced bubble wiring method for various materials”
Dear Reviewer
Thank you very much for your valuable comments.
We have revised our previous manuscript based on your provided comments. The revised points are follows.
1. We revised our title from “Plasma-induced bubble wiring method for various materials” to “Plasma-induced bubble microjet metallization of elastomer”. Thank you for your comment. Because we have only tried elastomers in this paper, we have modified the title as your comment. In addition, we adjusted the keyword of abstract from “plasma, microbubble, wiring” to “nickel, nanoparticles, wiring, adhesion, latex rubber”.
2. We have added a new section of “4.3 discussion” and compare some numerical properties of our method with the other developed method. I am afraid that it was difficult to measure the resistance value because the particle adhesion by the present method is low in density. This is due to that the current flowed by dielectric breakdown between the adhered particles, hence we have compared the wiring speed, heat treatment, resolution and total process time, instead. The investigation is under going and relationship between resistance value and thickness of nickel layer will be provided in next coming paper.
3. About figure 4, we have changed the image from 3D to 2D. Also, we agree the words “implant” and “embed” were inappropriate, therefore we changed these words to “adhere”. Thank you very much.
4. Thank you for your additional comment. We have added and revised following points. ①. The word “200%” was added in the section 4.1. This means we stretched the rubber twice during experiment. ②. The diameter of nanoparticle D was revised from 100 µm to 100 nm. ③. Figure 10 (revised version is figure 11) was redrawn in the electrical circuit schematic. In addition, we changed the word “non-inductive resister” to “resister”.
We hope to hear from you soon.
Sincerely,
Keita Ichikawa

Reviewer 2 Report
This work reported a plasma-induced bubble based method to deposit microparticles on material surfaces. This method utilized the micro-jet flow from a collapsing bubble to break target surface and implant particles. It avoided any surface treatments and was able to wire surface with any predefined patterns. I suggest to accept this work after addressing the following comments.
(1) The collapsing bubble "breaks" the surfaces. I am interested in how the surface looks after bubble treatment. It would be better to include any SEM photos of the surface before/after bubble treatments.
(2) Followed by the last comments, since this method relies on surface breakage, the wiring performance should depend on the stiffness and/or other mechanical modulus of the surface material. Please indicate which type of surface is more suitable for this wiring method.
(3) What is the forces between microparticles and surfaces? How stable is the wires? How does it resist to flow, scratch, etc.
(4) What is the maximum moving speed of the device? What is the maximum frequency to generate and break a bubble? These factors might determine the speed of drawing a wire.
Author Response
To
The editor and reviewers of Journal of micromachines
Subject: Submission of manuscript entitled “Plasma-induced bubble wiring method for various materials”
Dear Reviewer
Thank you very much for your valuable comments.
We revised our previous manuscript based on your provided comments. The revised points are as follows.
1. Thank you to your comment, and new SEM image was provided in figure 10.
As shown in Figure 10, we have successfully obtained the rubber surface, the cross-section, and the fault plane of the rubber.
2. Thank you for your comment about appropriate material for our technology. With regard to the adhesion between rubber and particles, we considered that electrostatic force was the most powerful. Therefore, we believe stretchable and easily chargeable materials are suitable for our approach.
3. Thank you for your comment. We performed peel test with sellotape and confirmed that the created line on the rubber never peel off by the striping of the sellotape for numerical times. Therefore, it is reasonable to say that the created line is stability as a general electrical circuit.
4. Thank you for your comment. We have added maximum moving speed of the device and maximum frequency to generate bubble in the section 4.2. Moreover, we compared these properties with the other wiring technology in the new section 4.3.
We hope to hear from you soon.
Sincerely,
Keita Ichikawa

Round 2
Reviewer 1 Report
One major comment:
Because you demonstrate 500 micrometer resolution, you cannot write "several tens of microns" in the abstract. You must write "500 micron resolution".
In the Discussion section 4.3 you mention that you now know how to improve resolution. I would formulate this as something like "we are currently working on bubble injector redesign which promises to improve resolution significantly". This way you do not give away the main result of your next paper.
Minor comments:
3.1: "diameter of the adhesion nickel particles" ==> diameter of the metallized area
4.1: "particle adhesion density" sounds strange. Particle density is enough.
4.2 "without peeling" is ambiguous because peeling has connotation as active procedure. I propose "withput delamination".
Fig. 11 (and caption): resister ==> resistor
Reviewer 2 Report
In the revised version, the authors have well addressed my comments, and provided further experimental details. I believe the concept of this work will benefit a wide field of applications. I suggest to accept as is.